# SARS-CoV-2 lineage B.6 was the major contributor to early pandemic transmission in Malaysia

Yoong Min Chong[1], I-Ching Sam[1,2]*, Jennifer Chong[2], Maria Kahar Bador[1,2], Sasheela Ponnampalavanar[3], Sharifah Faridah Syed Omar[3], Adeeba Kamarulzaman[3], Vijayan Munusamy[3], Chee Kuan Wong[3], Fadhil Hadi Jamaluddin[4], Yoke Fun Chan[1]*

1 Department of Medical Microbiology, Faculty of Medicine, University of Malaya, Kuala Lumpur, Malaysia, 2 Department of Medical Microbiology, University of Malaya Medical Centre, Kuala Lumpur, Malaysia, 3 Department of Medicine, Faculty of Medicine, University of Malaya, Kuala Lumpur, Malaysia, 4 Department of Anesthesiology, Faculty of Medicine, University of Malaya, Kuala Lumpur, Malaysia

* jicsam@ummc.edu.my (I-CS); chanyf@um.edu.my (YFC)

## Abstract

Malaysia had 10,219 confirmed cases of COVID-19 as of September 20, 2020. About 33% were associated with a Tablighi Jamaat religious mass gathering held in Kuala Lumpur between February 27 and March 3, 2020, which drove community transmission during Malaysia's second wave. We analysed genome sequences of SARS-CoV-2 from Malaysia to better understand the molecular epidemiology and spread. We obtained 58 SARS-CoV-2 whole genome sequences from patients in Kuala Lumpur and performed phylogenetic analyses on these and a further 57 Malaysian sequences available in the GISAID database. Nine different SARS-CoV-2 lineages (A, B, B.1, B.1.1, B.1.1.1, B.1.36, B.2, B.3 and B.6) were detected in Malaysia. The B.6 lineage was first reported a week after the Tablighi mass gathering and became predominant (65.2%) despite being relatively rare (1.4%) globally. Direct epidemiological links between lineage B.6 viruses and the mass gathering were identified. Increases in reported total cases, Tablighi-associated cases, and community-acquired B.6 lineage strains were temporally linked. Non-B.6 lineages were mainly travel-associated and showed limited onward transmission. There were also temporally correlated increases in B.6 sequences in other Southeast Asian countries, India and Australia, linked to participants returning from this event. Over 95% of global B.6 sequences originated from Asia Pacific. We also report a nsp3-C6310A substitution found in 47.3% of global B.6 sequences which was associated with reduced sensitivity using a commercial diagnostic real-time PCR assay. Lineage B.6 became the predominant cause of community transmission in Malaysia after likely introduction during a religious mass gathering. This event also contributed to spikes of lineage B.6 in other countries in the Asia-Pacific. Mass gatherings can be significant causes of local and global spread of COVID-19. Shared genomic surveillance can be used to identify SARS-CoV-2 transmission chains to aid prevention and control, and to monitor diagnostic molecular assays.

**Clinical Trial Registration**: COVID-19 paper.

**Data Availability Statement:** All relevant data are within the manuscript and its Supporting Information files.

**Funding:** Sam IC and Chan YF received grant (HDTRA1-17-1-0027) from Defense Threat Reduction Agency, USA. The funder had no role in study design, data collection and analysis, decision to publish, or preparation of the manuscript.

**Competing interests:** The authors have declared that no competing interests exist.

## Author summary

The early COVID-19 pandemic in Malaysia was driven mainly by transmission following a religious mass gathering held in Kuala Lumpur at the end of February 2020. To study the genetic epidemiology of SARS-CoV-2 in Malaysia, we analysed 57 available and 58 newly generated Malaysian whole genome virus sequences. We found that lineage B.6, rare (1.4%) globally, first appeared after the mass gathering, was linked to attendees, and became predominant (65.2%) in Malaysia. Increases in COVID-19 cases and locally acquired B.6 strains were temporally linked. Non-B.6 viruses were mainly associated with travel and showed limited spread. Increases in B.6 viruses in Asia Pacific countries were temporally linked to participants returning from this mass gathering. Altogether, 95% of global B.6 sequences originated in Asia Pacific countries. We also report a mutation in the virus nsp3 gene found in 47.3% of global B.6 sequences and associated with reduced detection by a commercial diagnostic test. In conclusion, the religious mass gathering in Kuala Lumpur was associated with the second wave of COVID-19 cases of predominantly B.6 lineage in Malaysia, and subsequent spread of B.6 viruses regionally. Genome sequence data provides valuable insight into virus spread and is important for monitoring continued accuracy of diagnostic kits.

## Introduction

Coronavirus disease (COVID-19), caused by severe acute respiratory syndrome coronavirus 2 (SARS-CoV-2), has caused more than 33 million infections and 1 million deaths globally [1]. In Malaysia, there have been 10,219 confirmed cases with 130 deaths reported as of September 20, 2020 [2], out of a total population of 32.7 million, giving a cumulative incidence of 31 per 100,000. COVID-19 was first identified in Malaysia on 25 January 2020 with early sporadic cases mainly associated with travel from China and Singapore. The second wave of infection began in early March 2020, and was associated with attendance at a religious mass gathering held by the Tablighi Jamaat group in Kuala Lumpur between 27 February to 3 March. This gathering was attended by 16,000 people, including about 1,500 from overseas, including Australia, Canada, Nigeria, and India, and neighbouring Southeast Asian countries Singapore, Indonesia, Thailand, Cambodia, Vietnam, Brunei and Philippines [3]. As the number of confirmed cases rapidly increased, Malaysia had the highest number of confirmed cases in Southeast Asia in March and early April. A movement control order, or lockdown, restricted movement except for necessity, work, and health circumstances from 18 March. Through rigorous public health measures, case numbers began to fall, allowing phased lifting of restrictions after 3 months.

SARS-CoV-2 is an RNA virus from the family of *Coronaviridae* and genus *Betacoronavirus*. There are currently more than 100,000 publicly available complete or near-complete genome sequences of SARS-CoV-2 with over 100 lineages identified. As the virus is rapidly evolving, it is important to understand the genomic epidemiology of SARS-CoV-2 to track virus evolution and local and global spread.

Here we report the whole genome sequences of SARS-CoV-2 from 58 patients from University Malaysia Medical Centre (UMMC), a designated COVID-19 hospital in Kuala Lumpur. Our objective was to study the genetic diversity and epidemiology of SARS-CoV-2 in Malaysia, relative to the regional and worldwide virus lineages. In particular, we were interested in determining the possible role of the Tablighi Jamaat gathering in national and regional spread.

Genomic surveillance improves monitoring and understanding of current SARS-CoV-2 spread within a country and can help inform public health measures.

## Methods

### Ethics statement

This study was approved by the University Malaya Medical Centre ethics committee (no. 2020730–8928). Our institution does not require informed consent for retrospective studies of archived and anonymised samples.

### Clinical sample collection

Patients admitted to UMMC with suspected COVID-19 were diagnosed by real-time PCR detection of SARS-CoV-2 in nasopharyngeal and oropharyngeal swabs, using a WHO-recommended Berlin Charité protocol [4] and commercial assays. For each patient, the earliest available sample during illness with the highest viral load was selected for direct sequencing. Cases were diagnosed between February and April 2020.

### Whole genome sequencing analysis

Viral RNA was extracted from 58 positive clinical samples using QIAamp Viral RNA Mini Kit (Qiagen, Germany) and amplified according to the ARTIC-nCoV-2019 protocol [5]. Briefly, cDNA was synthesized using SuperScript IV First-Strand Synthesis System (Invitrogen, USA) with random hexamers. Multiplex PCR was performed with Q5 High-Fidelity DNA Polymerase (NEB, USA) using two pools of 109 nCoV-2019/V3 primer sets to generate overlapping 400 nucleotide amplicons. PCR products were pooled into one tube and purified using AMPure XP beads (Beckman Coulter, USA). The iTrue library method was used for library preparation [6]. Libraries were sequenced using iSeq 100 reagent kit (Illumina, USA) on the iSeq 100 system (Illumina, USA), with output of $1 \times 300bp$ reads.

### Bioinformatic analysis

The sequenced reads were analysed using Geneious Prime 2020 (Biomatters, New Zealand). Reads were trimmed for quality using default parameters and mapped to reference strain Wuhan-Hu-1 (GenBank accession number MN908947). Leftover gaps were sequenced separately by conventional Sanger sequencing using the nCoV-2019/V3 primers set and consensus sequences were generated. Multiple sequence alignment was performed using MAFFT with default parameters [7]. Phylogenetic analysis was conducted with RAxML 8.2.11 implemented in Geneious with default parameters (generalized time-reversal (GTR) + gamma substitution model and bootstrapped 1000 times). The analysed Malaysian sequences comprised the 58 samples from this study and 57 other Malaysian genome sequences available at GISAID (www. gisaid.org) as of September 20, 2020 [8], which include 4 previously published by our centre [9]. Therefore, 62 of the Malaysian sequences were from our centre. Variant analysis was performed among all Malaysian sequences using Geneious. Unique nucleotide mutations were further confirmed using the CoV-GLUE online tool (http://cov-glue.cvr.gla.ac.uk) [10]. Lineage groups were classified according to the Pangolin COVID-19 Lineage Assigner online tool (www.pangolin.cog-uk.io) [11].

SARS-CoV-2 lineage B.6 complete genome sequences up to September 20, 2020 were retrieved from GISAID. A total of 1,497 B.6 sequences, including 75 Malaysian B.6 sequences (43 from our study) were available in the GISAID database. Of these, 1,122 high quality B.6 sequences with > 29,000 bp, < 1% Ns and < 0.05% unique amino acid mutations were aligned

using MAFFT with default parameters in Geneious. The alignment was then subjected to maximum-likelihood (ML) phylogenetic analyses using IQTREE v1.6.12 [12], using the GTR+F+I +G4 nucleotide substitution model and assessing branch support by the Shimodaira-Hasegawa-like approximate likelihood ratio test with 1,000 replicates. The ML tree was visualized using FigTree v1.4 [13].

## Results

Fifty-eight whole genome sequences with >99% reads mapped to the reference genome were generated, with average coverage depth of 944× (range, 74× to 7119×; S1 Table). The consensus sequences have been deposited in the GISAID database (accession numbers EPI_ISL_501176 to EPI_ISL_501228, and EPI_ISL_506996 to EPI_ISL_50700). All other Malaysian whole genome sequences (57 sequences) available in GISAID were added to the analysis, making a total of 115 Malaysian sequences.

Nine lineage groups were identified in the 115 Malaysian sequences: A, B, B.1, B.1.1, B.1.1.1, B.1.36, B.2, B.3 and B.6 (Fig 1). The most identified lineage group was B.6 (n = 75, 65.2%), followed by B (n = 21, 18.3%), B.1.1 (n = 5, 4.4%), B.1 (n = 4, 3.5%), A (n = 3, 2.6%), B.1.1.1 (n = 2, 1.7%), B.2 (n = 2, 1.7%), B.1.36 (n = 2, 1.7%) and B.3 (n = 1, 0.9%).

Although lineage B.6 (which also belongs to GISAID clade O and Nextstrain clade 19A) was the most common in Malaysia, only 1,497 B.6 sequences had been reported globally out of over 105,568 sequences available as of September 20, 2020, a rate of 1.4%. Even after considering 17 sequences generated from the current study which were part of a healthcare-associated cluster, B.6 remained the most frequently detected lineage in Malaysia. Of the 62 cases from our centre, comprising 58 from this study and 4 from an earlier study [9], 7 had direct links to the Tablighi Jamaat gathering– 4 had attended and 3 were contacts of attendees. All 7 had B.6 lineage viruses, thus establishing a link between the Tablighi Jamaat and the B.6 lineage.

The Tablighi Jamaat lasted from February 27 to March 3, 2020, and the first reported B.6 sequences globally were from Malaysia, Philippines and Taiwan on March 4 (Fig 2A). Within a week of the Tablighi Jamaat ending, there was a large increase in reported COVID-19 cases in Malaysia, with the vast majority being associated with this gathering. This was accompanied by a spike of lineage B.6 in Malaysia and other countries (Fig 2A and 2B). From March 3 onwards, lineage B.6 sequences comprised 46/61 (75.4%) of sequences from our centre and 29/39 (74.4%) of sequences submitted by other Malaysian centres, indicating that this finding was not specific to our centre. This increase in cases led to the imposition of a nationwide movement control order on March 18.

Of the 1,497 globally reported B.6 sequences, 1,422 (95%) originated from Asia Pacific countries. Singapore had the highest number of sequences (664), followed by India (532), Australia (81), Malaysia (75), USA (25), Philippines (16), Timor-Leste (14), Canada (13), United Kingdom (10), Thailand (9), Taiwan (7), Brunei (5) and Israel (5). In several Southeast Asian countries (Malaysia, Singapore, Brunei, Timor-Leste, Philippines and Myanmar), lineage B.6 became predominant and accounted for 50–100% of total sequences in GISAID (Fig 2C). Of the 7 Malaysian B.6 cases with Tablighi links, sequence 188407 and 3998 clustered with sequences from Singapore, India and Australia; sequences 2063 and 5096 clustered with sequences from Singapore; sequence 2875 grouped with sequences from Singapore, India and USA; sequence 2297 linked with Singapore, India and New Zealand; and sequence 1399 clustered with those from India, USA, Oman, Australia, Taiwan, Canada, Guam and Sierra Leone (Fig 3). The appearance and surge of overall COVID-19 cases, Tablighi-associated cases, and B.6 lineage cases in Malaysia were thus temporally linked, as were B.6 lineage cases in other Asia Pacific countries.

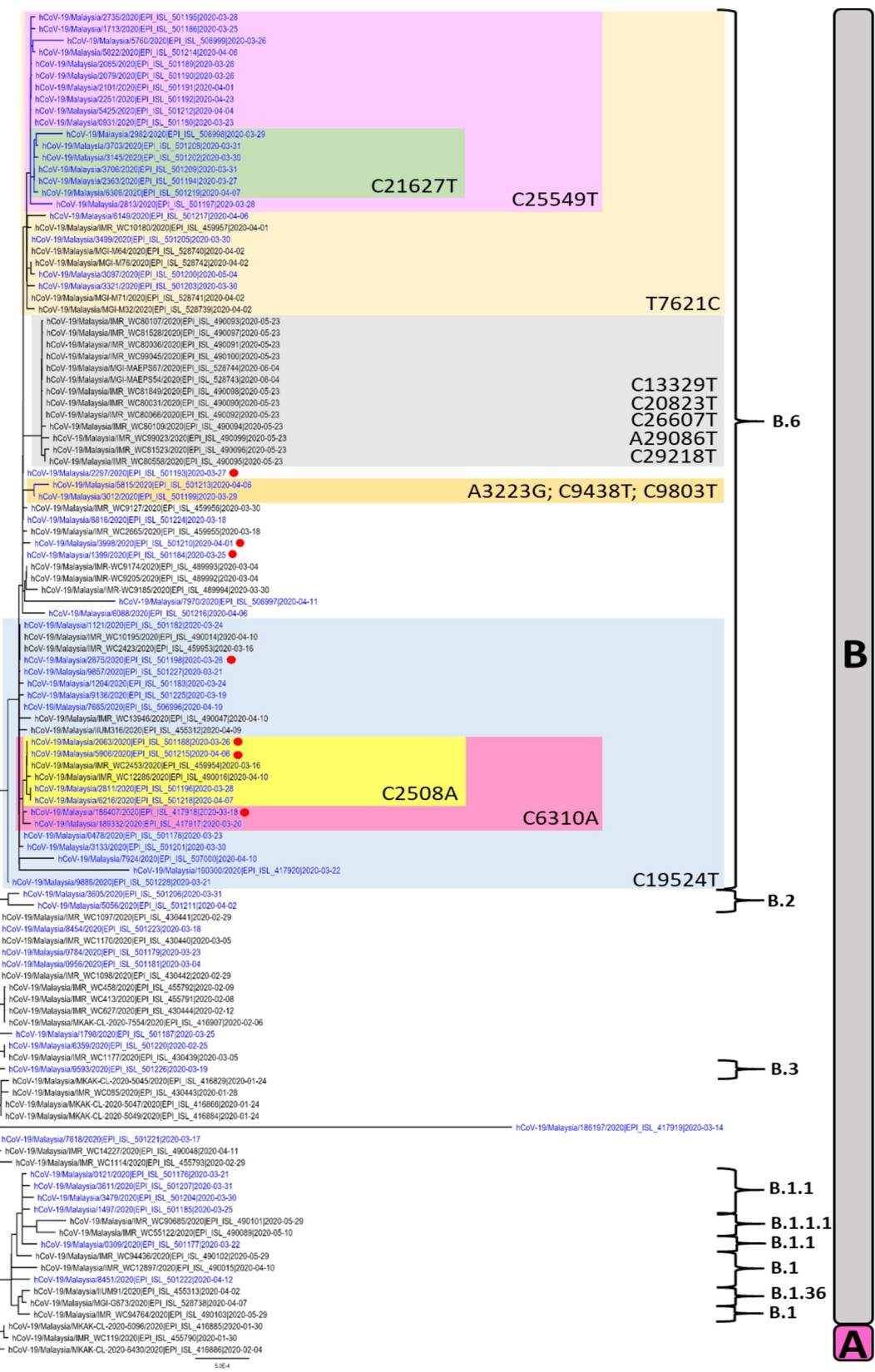

**Fig 1. Phylogenetic tree of 115 SARS-CoV-2 whole genome sequences from Malaysia.** Sequences from our study are shown in blue and the 7 sequences with known links to the Tablighi Jamaat gathering are denoted with red circles. The tree is rooted on the branch separating lineages A and B. The key single nucleotide variants in lineage B.6 are highlighted.

Several distinct mutations were observed among the Malaysian B.6 sequences. The 17 sequences in the healthcare-associated cluster had a non-synonymous mutation C25549T (P53F) in ORF3a (Fig 1). A secondary cluster within this healthcare-associated cluster contained six sequences with an additional mutation C21627T (T22I) in the spike protein. Eight Malaysian sequences, including 6 from this study, had the additional substitution C6310A, resulting in the amino acid change S1197R in nsp3. This C6310A mutation led to reduced sensitivity by 1,000–10,000 viral copies ($>$ 10 cycles) with one of the commercial real-time PCR assays used in our laboratory when compared to the well-established Charité assay [4] and updated primers/probes issued by the assay manufacturer in response to this likely sequence mismatch (S2 Table). Some mutations were unique to Malaysian B.6 genomes, such as C2508A, A3223G, C9438T, C9803T, C13329T, C20823T, C26607T, A29086T and C29218T; others, including C6310A, T7621C and C19524T, were also observed in B.6 sequences from other countries including Singapore, Australia and India (Fig 3).

The mutation D614G in the spike protein, which may increase infectivity of SARS-CoV-2 [14] and has become prevalent in many countries, was only observed in 13 (11.3%) Malaysian sequences from lineages B.1, B.1.1, B.1.1.1 and B.1.36 (S3 Table), but was not present in lineage B.6. None of the 41 rare receptor binding variants described in GISAID–including the two most common, S477N and N439K –were observed in the Malaysian sequences.

## Discussion

In Malaysia, the Tablighi Jamaat mass gathering became the largest COVID-19 cluster, resulting in the screening of over 42,000 individuals, with 3,375 confirmed cases (33% of national cases) and 34 deaths, of which 29 (85.3%) were aged 50 and above. Multiple generations and at least 17 sub-clusters linked to this gathering were detected [14]. Of the cases, 825 (24.4%) were foreign nationals from 28 countries, including Southeast Asian countries, Australia, India, Pakistan and Bangladesh [15]. Religious gatherings were recognized early in the pandemic as a significant factor in the spread of COVID-19 [16], with large clusters associated with events in India, Iran and South Korea [17–20].

We established a direct epidemiological link between the Tablighi Jamaat event and lineage B.6 viruses for 7 cases. The first reported B.6 viruses date from a week after the Tablighi Jamaat event started. Thus, the appearance of the B.6 lineage, and subsequent surge of overall COVID-19 cases, Tablighi-associated cases, and B.6 lineage cases in Malaysia were temporally linked, as were B.6 lineage cases in other Asia Pacific countries. The diversity of the 7 Tablighi-linked Malaysian B.6 sequences (Fig 3) suggests there may have been multiple B.6 strains involved in the initial exposure, which is compatible with the large number of attendees from many different countries who could have been the sources. The participants spent several days in large indoor congregations with shared eating and sleeping arrangements [3], which would have led to extensive transmissions. After the event, non-Kuala Lumpur residents (Malaysian and foreign) would have returned to their areas of residence.

Attendees at the Kuala Lumpur event also apparently acted as sources of further extensive transmission occurring at two subsequent Tablighi Jamaat gathering in Pakistan and India later in March [21]. This may explain a later rise in B.6 in India in late March (Fig 2B). The first known COVID-19 case in Brunei returned from the Kuala Lumpur mass gathering and 52.6% of subsequently reported Brunei cases were associated with this event as of early April

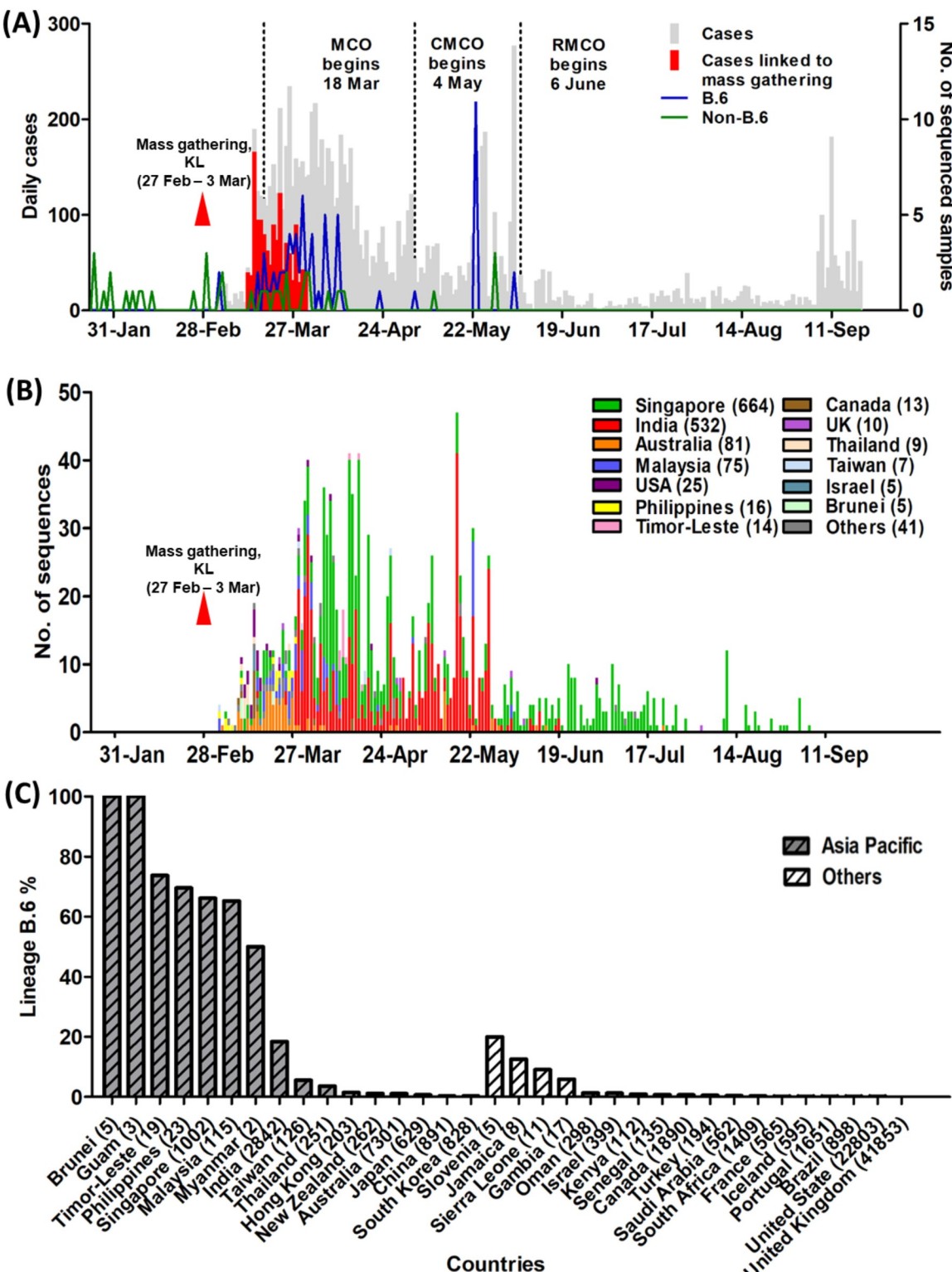

**Fig 2. (A) Epidemic curve of COVID-19 in Malaysia and time course of reported total cases, Tablighi-associated cases, and lineages based on 115 Malaysian sequences, from the first reported case on 25 January to 20 September 2020.** The Tablighi Jamaat gathering in Kuala Lumpur is labelled as "mass gathering, KL". The different phases of lockdown are shown, comprising the movement control order (MCO), conditional movement control order (CMCO) and recovery movement control order (RMCO). **(B) Time course of global reported lineage B.6 SARS-CoV-2 sequences available in the GISAID database (as of 20 September 2020).** The timescale of

the horizontal axis is the same as (A). Countries with fewer than 5 lineage B.6 sequences are included in "others", comprising Brazil (1), China (3), France (1), Gambia (1), Guam (3), Hong Kong (1), Iceland (1), Jamaica (1), Japan (4), Kenya (1), Myanmar (1), New Zealand (3), Oman (4), Portugal (2), Saudi Arabia (2), Senegal (1), Sierra Leone (1), South Africa (4), South Korea (2), Slovenia (1) and Turkey (1). **(C) The percentage rates of lineage B.6 sequences over total available complete sequences (in brackets) from each country available at GISAID as of 20 September 2020.** Asia Pacific countries are colored grey.

[22]. Southern Thailand also experienced an outbreak of COVID-19 after Thai participants returning from this event had a reported 29.3% attack rate [23]. Other Southeast Asian countries including Cambodia, Indonesia, Singapore, Philippines and Vietnam also reported imported cases associated with this gathering [24–28]. A caveat is that the number of whole genome sequences reported from developing countries in Asia is relatively low and likely underreports the true incidence of B.6. However, it is notable that this lineage is a very minor contributor in developed countries with greater sequencing efforts (Fig 3C). Detailed, publicly available epidemiological data are also not available for most of the B.6 sequences outside our centre, and would be very useful to further support the involvement of Tablighi Jamaat participants.

Whole genome sequencing of SARS-CoV-2 in our centre not only allowed important insights into local and regional epidemiology, but was also used to track networks within a healthcare-associated cluster, which will be reported elsewhere. Furthermore, public sharing of sequences contributed to identification of the C6310A (nsp3-S1197R) mutation affecting the sensitivity of a commercial diagnostic real-time PCR assay, leading to updated primers/probes. This mutation is present in only 0.7% of global sequences, but as many as 708 (47.3%) of 1,497 available lineage B.6 sequences, and within our centre we estimate that the reduced sensitivity of the original assay impacted 10% of positive samples. With many diagnostic kits now on the market and the continued evolution of the virus, this shows the importance of choosing reputable manufacturers who are open about the primer and probes used [29] and diligently monitor new circulating sequences for potential primer/probe mismatches [30]. Assays targeting more than one gene are also preferable [29].

There were 40 (34.8%) non-B.6 lineage sequences reported from Malaysia, including 15 from our study. At least 25 (62.5%) reported recent international travel (including 3 of the earliest cases from lineage A, imported from China), compared to just 2/46 (4.3%) of our centre's B.6 cases. Our sequenced cases with lineages B.1.1, B.2 and B.3 had travelled from Europe and a single B.1 case had returned from a cruise to the Americas. The B.1 and B.1.1 lineages are the most common in Americas and Europe [31]. The small numbers of other non-B.6 lineages indicate multiple introductions from overseas that failed to establish significant transmission in Malaysia, as the movement control order from 18 March closed the country's borders and mandated quarantine for all incoming travellers. However, by this point the B.6 lineage had established community transmission, reflected by the local acquisition of disease in most of our B.6 cases without direct or discernible links to the Tablighi Jamaat event.

In summary, initial COVID-19 spread and the predominance of lineage B.6 were temporally and epidemiologically associated with the Tablighi Jamaat religious gathering, and led to established community spread during the second wave in Malaysia. Attendees at this event likely spread strains of the B.6 lineage to other countries in the region, including Southeast Asian countries, India and Australia. Further sequence and epidemiological data are needed from Malaysia and other countries in the region to understand the full extent of the spread of B.6. The use of genomic data is important to rapidly identify possible transmission chains and provide a framework for the response to COVID-19.

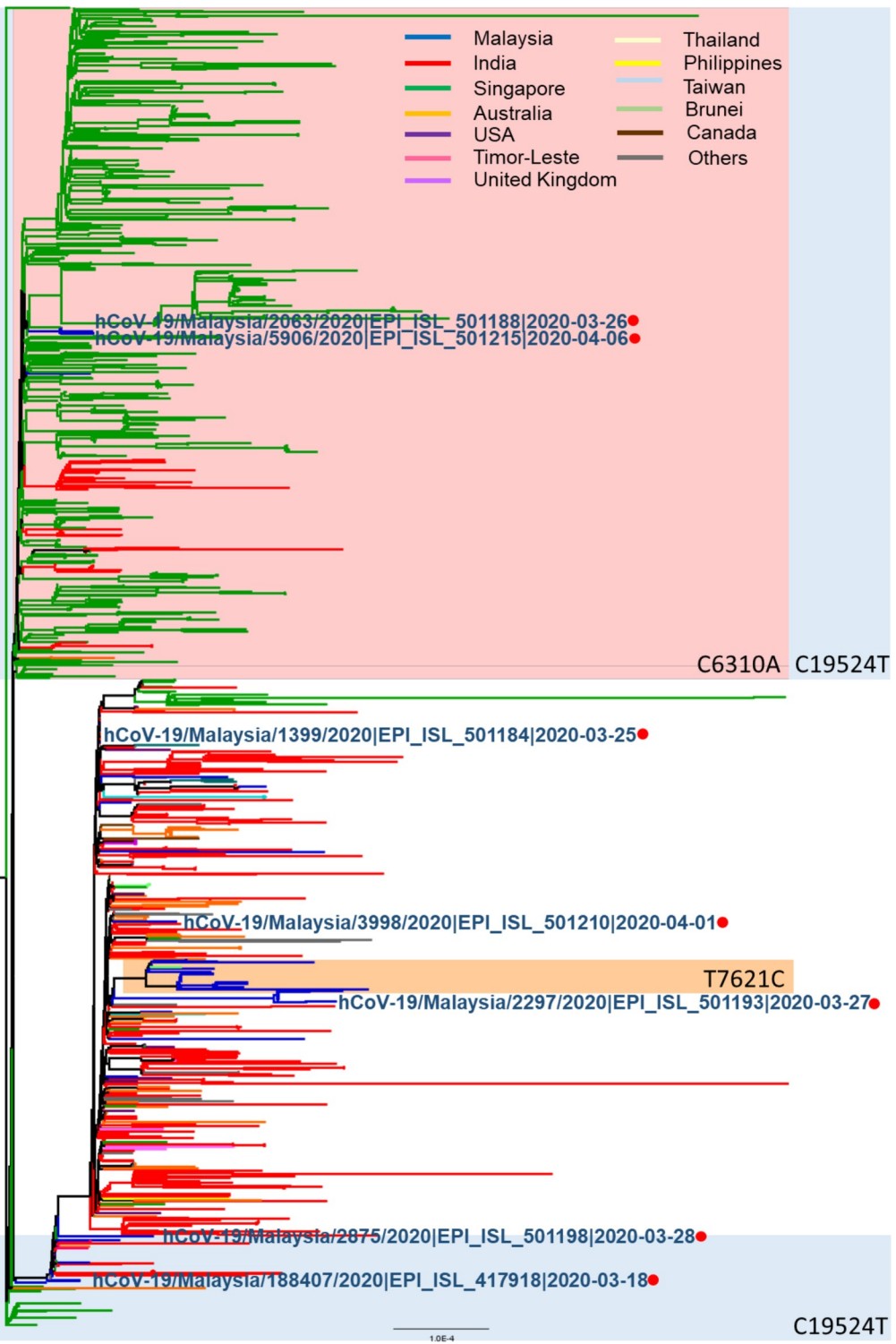

**Fig 3. Phylogenetic tree of 1,122 high-quality B.6 lineage SARS-CoV-2 sequences with > 29,000 bp, < 1% Ns and < 0.05% unique amino acid mutations available from the GISAID database as of 20 September 2020.** Sequences linked to the Tablighi Jamaat gathering are labelled in blue and denoted with red circles.

## Supporting information

**S1 Table. List of 115 SARS-CoV-2 genomes derived from Malaysian samples and available in GISAID which were used in this study.** The first 58 sequences were generated in this study. Sequences 77–80 were published by our centre [9].
(DOCX)

**S2 Table. Comparison of threshold cycle (Ct) values obtained using the Berlin Charité RT-qPCR assay (E gene) [4] and commercial RT-qPCR assay (nsp3 gene; original and updated primers/probes) using 16 patient samples from this or a previous study [9] categorised by presence/absence of the nsp3-C6310A substitution.** A difference in viral RNA of one log or 10× in the initial template concentration is equivalent to a Ct difference of 3.3.
(DOCX)

**S3 Table. Nucleotide and amino acid changes in different SARS-CoV-2 lineages in the 115 Malaysian sequences compared to the reference strain Wuhan-Hu-1 (MN908947).** Amino acids are denoted in parentheses and unique mutations found in each lineage are bold and highlighted in grey. The sequence names are colored according to their lineage: Lineage A in red color, lineage B in black color, lineage B.1 in purple color, lineage B.1.1 in pink color, B.1.1.1 in golden color, B.1.36 in brown color, B.2 in orange color, B.3 in green color and B.6 in blue color.
(XLSX)

## Acknowledgments

We gratefully acknowledge the authors from originating and submitting laboratories of GISAID sequence data on which the analysis is based. We thank GenSeq Sdn Bhd, Malaysia for their assistance in sequencing. The authors are part of the University Malaya COVID-19 Research Group, which include the healthcare workers involved in care of COVID-19 patients in the University of Malaya Medical Centre.

## Author Contributions

**Conceptualization:** Yoong Min Chong, I-Ching Sam, Yoke Fun Chan.

**Data curation:** Yoong Min Chong, I-Ching Sam, Yoke Fun Chan.

**Formal analysis:** Yoong Min Chong, I-Ching Sam, Yoke Fun Chan.

**Funding acquisition:** I-Ching Sam, Yoke Fun Chan.

**Investigation:** Yoong Min Chong, I-Ching Sam, Jennifer Chong, Maria Kahar Bador, Sasheela Ponnampalavanar, Sharifah Faridah Syed Omar, Adeeba Kamarulzaman, Vijayan Munusamy, Chee Kuan Wong, Fadhil Hadi Jamaluddin, Yoke Fun Chan.

**Methodology:** Yoong Min Chong.

**Project administration:** Yoong Min Chong, I-Ching Sam, Yoke Fun Chan.

**Resources:** Yoong Min Chong, I-Ching Sam, Yoke Fun Chan.

**Software:** Yoong Min Chong, I-Ching Sam, Yoke Fun Chan.

**Supervision:** I-Ching Sam, Yoke Fun Chan.

**Validation:** I-Ching Sam, Yoke Fun Chan.

**Visualization:** I-Ching Sam, Yoke Fun Chan.

**Writing – original draft:** Yoong Min Chong, I-Ching Sam, Yoke Fun Chan.

**Writing – review & editing:** Yoong Min Chong, I-Ching Sam, Adeeba Kamarulzaman, Yoke Fun Chan.

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
