## [Decision Letter · Decision Letter 0]

13 Sep 2020

Dear Dr Chan,

Thank you very much for submitting your manuscript "SARS-CoV-2 lineage B.6 is the major contributor to transmission in Malaysia" for consideration at PLOS Neglected Tropical Diseases. As with all papers reviewed by the journal, your manuscript was reviewed by members of the editorial board and by several independent reviewers. In light of the reviews (below this email), we would like to invite the resubmission of a significantly-revised version that takes into account the reviewers' comments. 

Your manuscript has been reviewed by three experts in the field. Reviewers have made some major suggestions for improvement. Please address all comments from all reviewers. In particular, you are strongly advised to perform in depth analyses of other genomic regions of the SARS-CoV-2 from different lineages and highlight the importance of this study towards the pandemic as suggested by reviewers. 

I am willing to consider your revised manuscript that addresses all the suggestions and criticisms of the reviewer. The modified version of your paper may be sent back to the original reviewers prior to its possible acceptance. We cannot, of course, promise publication at that time.

We cannot make any decision about publication until we have seen the revised manuscript and your response to the reviewers' comments. Your revised manuscript is also likely to be sent to reviewers for further evaluation.

Sincerely,

Susanna Kar Pui Lau, M.D.

Deputy Editor

Susanna Kar Pui Lau

Deputy Editor

Your manuscript has been reviewed by three experts in the field. Reviewers have made some major suggestions for improvement. Please address all comments from all reviewers. In particular, you are strongly advised to perform in depth analyses of other genomic regions of the SARS-CoV-2 from different lineages and highlight the importance of this study towards the pandemic as suggested by reviewers. 

I am willing to consider your revised manuscript that addresses all the suggestions and criticisms of the reviewer. The modified version of your paper may be sent back to the original reviewers prior to its possible acceptance. We cannot, of course, promise publication at that time.

Reviewer's Responses to Questions

**Key Review Criteria Required for Acceptance?**

**Methods**

-Are the objectives of the study clearly articulated with a clear testable hypothesis stated?

-Is the study design appropriate to address the stated objectives?

-Is the population clearly described and appropriate for the hypothesis being tested?

-Is the sample size sufficient to ensure adequate power to address the hypothesis being tested?

-Were correct statistical analysis used to support conclusions?

-Are there concerns about ethical or regulatory requirements being met?

Reviewer #1: It is a SARS-CoV-2 retrospective genomic epidemiological study and the authors appear to have taken necessary steps to elucidate their findings as such.

Some clarification/changes regarding the sample sizes is required.

Reviewer #2: The objective of the study is clearly presented, the study design is appropriate, the population has been described. However, the sample size is limited compared to the confirmed cases. There are no ethical concerns. No particular statistics analysis have been used.

Reviewer #3: -Are the objectives of the study clearly articulated with a clear testable hypothesis stated?

-Is the study design appropriate to address the stated objectives?

-Is the population clearly described and appropriate for the hypothesis being tested?

-Is the sample size sufficient to ensure adequate power to address the hypothesis being tested?

-Were correct statistical analysis used to support conclusions?

-Are there concerns about ethical or regulatory requirements being met?

Yes

**Results**

-Does the analysis presented match the analysis plan?

-Are the results clearly and completely presented?

-Are the figures (Tables, Images) of sufficient quality for clarity?

Reviewer #1: The results require some clarification/changes. There are no major errors in the analysis.

Reviewer #2: The analysis matches the plan. The results are missing some information and also a new schematic chart would be helpful to better understand where the mutations are located.

Reviewer #3: YES

**Conclusions**

-Are the conclusions supported by the data presented?

-Are the limitations of analysis clearly described?

-Do the authors discuss how these data can be helpful to advance our understanding of the topic under study?

-Is public health relevance addressed?

Reviewer #1: The conclusions need some clarification/changes and there is acknowledgement of the limitations of the sequencing studies.

Reviewer #2: No limitations are mentioned. The conclusions reflect the data. However, the authors should highlight the importance of this data in the pandemic, how these data advance the global understanding of SARS-CoV-2 mechanisms of infections, spreading and lethality.

Reviewer #3: YES

**Editorial and Data Presentation Modifications?**

Reviewer #1: (No Response)

Reviewer #2: A new schematic chart would be helpful to better understand where the mutations are located. 

With this new chart specific for the mutations, the current trees should be changed.

Reviewer #3: NONE

**Summary and General Comments**

Reviewer #1: The authors describe a SARS-CoV-2 genomic epidemiology study that took place in Malaysia and the finding of a specific viral lineage that is circulating in SEA regions.

They also find that a certain mutation leads to reduced sensitivity in commercial assays for the virus.

Thanks to the authors to provide all necessary details to gauge the sequencing and analysis protocols used in the study.

It would be very helpful if the significance of the study to researchers and clinicians/diagnostics can be described in greater detail (Abstract and Introduction).

"In particular, we were interested in determining the role of the Tablighi Jamaat gathering in regional spread."

Were the 58 patients from the religious gathering?

In the Methodology, explicit details are needed on how many of the 58 patients were from the religious gathering in Malaysia.

And how many of the other 50 Malaysian sequences were from the religious gathering? (Is it 39?)

The total number of B.6 sequences is not consistent between the last paragraph of Methodology and 3rd paragraph of Results.

A detailed split of B.6 sequences can be provided as: sequenced at UMMC + sequenced at other centers in Malaysia + sequenced at other global centers.

Regarding the events that took place and the statement, "of other Malaysian sequences in GISAID dating from the Tablighi Jamaat gathering".

In the above statement, the timeline of events are not clear.

Also, in the recently published work of the authors "Complete Genome Sequences of SARS-CoV-2 Strains Detected in Malaysia",

Fig. 1A the religious event is dated as 29 Feb - 3 Mar, but in the current study the date is 27 Feb - 1 Mar.

Thus, the authors should describe with detail and certainty about the events that took place and that the sequences were from infections originating at the religious gathering.

Rearranging some final sections of the Methodology and beginning of Results could be helpful.

Fig.1: If there are 39 sequences from the religious gathering why are only 3 highlighted?

To avoid blocking the dendrogram, moving the "red point" to the right side of the sequences would be better.

Fig. 3A: Onlyan outline of the histogram could be sufficient. This would also make the lineage markers more clear to see.

Fig. 3C: Would be helpful to include legend in the figure for Southeast Asian country bars vs other country bars

Reviewer #2: The manuscript “SARS-CoV-2 lineage B.6 is the major contributor to transmission in Malaysia” describes the predominant spreading of the B.6 lineage in Malaysia after a religious mass gathering in Kuala Lumpur. This event is associated with the main wave of COVID-19 cases in Malaysia. The authors analyzed 58 genomes sequences from COVID-19 patients that attended the religious event in Kuala Lumpur and 50 Malaysian sequences available in the GISAID database. The authors detected 9 different lineages, and among these, the B.6 became the predominant cause of community transmission in Malaysia after the likely introduction during the religious mass gathering.

The references are appropriate. 

Major revisions: 

- The authors use sometimes SARS-CoV-2 and COVID-19 as synonymous. They are not: SARS-CoV-2 is the novel coronavirus, while COVID-19 includes all the medical conditions caused by SARS-CoV-2 (as also correctly stated by the authors in the introduction). I would suggest the authors to review the manuscript and use the correct terminology.

- The authors should consider adding a more in-depth analysis of the sequences and their mutations. The authors just mentioned a mutation in ORF3a, one in the spike protein, and few more. What about the other viral genes? For example, the one in the polymerase mentioned by Pachetti et al, J Transl Med 18, 179 (2020)? 

- A schematic chart would be helpful to better understand where the mutations are located. 

- A few typos need to be corrected by the authors.

Reviewer #3: This is a well written manuscript demonstrating how genetic analysis can inform about transmission dynamics of SARS-CoV-2.

The study shows very nicely how a religious gathering fuelled the spread of the B.6 lineage in Malaysia. The results are clearly presented and my only comment to the results section is Fig. 3A where I would prefer to see the results as rates for instance per 100 000 population and not actual number of cases.

In the discussion I wonder if the authors could elaborate on whether the B.6 had a higher or lower mortality compared to other lineages?

PLOS authors have the option to publish the peer review history of their article (what does this mean?). If published, this will include your full peer review and any attached files.

Reviewer #1: No

Reviewer #2: No

Reviewer #3: No
---

## [Decision Letter · Decision Letter 1]

30 Oct 2020

Dear Dr Chan,

We are pleased to inform you that your manuscript 'SARS-CoV-2 lineage B.6 is the major contributor to transmission in Malaysia' has been provisionally accepted for publication in PLOS Neglected Tropical Diseases.

Best regards,

Susanna Kar Pui Lau, M.D.

Deputy Editor

Susanna Kar Pui Lau

Deputy Editor

Reviewer's Responses to Questions

**Key Review Criteria Required for Acceptance?**

**Methods**

-Are the objectives of the study clearly articulated with a clear testable hypothesis stated?

-Is the study design appropriate to address the stated objectives?

-Is the population clearly described and appropriate for the hypothesis being tested?

-Is the sample size sufficient to ensure adequate power to address the hypothesis being tested?

-Were correct statistical analysis used to support conclusions?

-Are there concerns about ethical or regulatory requirements being met?

Reviewer #1: This is a revised manuscript.

Reviewer #2: The authors updated the numbers and figures based on the data available up to September 20th 2020. The authors have taken the necessary steps to describe their findings.

There are no ethical concerns.

**Results**

-Does the analysis presented match the analysis plan?

-Are the results clearly and completely presented?

-Are the figures (Tables, Images) of sufficient quality for clarity?

Reviewer #1: (No Response)

Reviewer #2: The results have been better clarified. Table S3 has been added.

There are no major errors in the analysis.

**Conclusions**

-Are the conclusions supported by the data presented?

-Are the limitations of analysis clearly described?

-Do the authors discuss how these data can be helpful to advance our understanding of the topic under study?

-Is public health relevance addressed?

Reviewer #1: (No Response)

Reviewer #2: The conclusions have been improved and clarified.

**Editorial and Data Presentation Modifications?**

Reviewer #1: (No Response)

Reviewer #2: (No Response)

**Summary and General Comments**

Reviewer #1: The authors have addressed all the comments sufficiently.

Reviewer #2: The authors provided more details regarding the samples and the results.

A more in deep analysis of the sequences has been conducted.

minor revisions:

- the legends for figure 1-2-3 are missing.

PLOS authors have the option to publish the peer review history of their article (what does this mean?). If published, this will include your full peer review and any attached files.

Reviewer #1: No

Reviewer #2: No

---

## [Editor Report · Acceptance letter]

20 Nov 2020

Dear Dr Chan,

We are delighted to inform you that your manuscript, "SARS-CoV-2 lineage B.6 is the major contributor to early pandemic transmission in Malaysia," has been formally accepted for publication in PLOS Neglected Tropical Diseases.

Best regards,

Shaden Kamhawi

co-Editor-in-Chief

Paul Brindley

co-Editor-in-Chief
